# Application of 2D NMR Spectroscopy in Combination with Chemometric Tools for Classification of Natural Lignins

**DOI:** 10.3390/ijms241512403

**Published:** 2023-08-03

**Authors:** Anna V. Faleva, Ilya A. Grishanovich, Nikolay V. Ul’yanovskii, Dmitry S. Kosyakov

**Affiliations:** Laboratory of Natural Compounds Chemistry and Bioanalytics, Core Facility Center “Arktika”, M.V. Lomonosov Northern (Arctic) Federal University, Northern Dvina Emb. 17, 163002 Arkhangelsk, Russia; grilandr@gmail.com (I.A.G.); n.ulyanovsky@narfu.ru (N.V.U.)

**Keywords:** lignin, 2D NMR, HSQC, principal component analysis, hierarchical cluster analysis, classification

## Abstract

Lignin is considered a promising renewable source of valuable chemical compounds and a feedstock for the production of various materials. Its suitability for certain directions of processing is determined by the chemical structure of its macromolecules. Its formation depends on botanical origin, isolation procedure and other factors. Due to the complexity of the chemical composition, revealing the structural differences between lignins of various origins is a challenging task and requires the use of the most informative methods for obtaining and processing data. In the present study, a combination of two-dimensional nuclear magnetic resonance (2D NMR) spectroscopy and multivariate analysis of heteronuclear single quantum coherence (HSQC) spectra is proposed. Principal component analysis and hierarchical cluster analysis techniques demonstrated the possibility to effectively classify lignins at the level of belonging to classes and families of plants, and in some cases individual species, with an error rate for data classification of 2.3%. The reverse transformation of loading plots into the corresponding HSQC loading spectra allowed for structural information to be obtained about the latent components of lignins and their structural fragments (biomarkers) responsible for certain differences. As a result of the analysis of 34 coniferous, deciduous, and herbaceous lignins, 10 groups of key substructures were established. In addition to syringyl, guaiacyl, and *p*-hydroxyphenyl monomeric units, they include various terminal substructures: dihydroconiferyl alcohol, balanopholin, cinnamic acids, and tricin. It was shown that, in some cases, the substructures formed during the partial destruction of biopolymer macromolecules also have a significant effect on the classification of lignins of various origins.

## 1. Introduction

Being the second most abundant biopolymer in nature, lignin has attracted increasing attention from researchers as a renewable source of valuable chemical compounds and a feedstock for the production of various materials. Physical-chemical properties of lignin and thus its suitability for certain directions of processing are determined by the chemical structure of its macromolecules. They are formed as a result of enzymatic dehydrogenative polymerization of *p*-coumaryl, coniferyl, and synapyl alcohols (monolignols). The latter act as precursors for *p*-hydroxyphenyl (H), guaiacyl (G), and syringyl (S) phenylpropane structural units (PPU), respectively. An absence of genetic control of the polymerization process and a great diversity of bonds between the structural units of the macromolecule determine the irregularity of the polymer chains and the dependence of the resulting structure on many factors. Despite the fact that there are a large number of publications devoted to the structure of lignins of various origins [1,2,3,4,5], many aspects in this field still remain unclear due to its exceptional complexity. Naturally, the structure of lignin is primarily determined by the botanical origin of the biopolymer [5], but it also depends on the procedure used for isolating lignin preparations from the plant material [1,6,7,8,9]. At the same time, a number of recent studies showed that lignin structure can be additionally complicated due to the incorporation of non-typical moieties, such as catechin-type phenylpropane units (C) [4] and flavonoids (for example, tricin) [5,10]. 

In this regard, a detailed classification of lignins should be based on the most complete consideration of both major and minor structural features of preparations obtained from various plant materials. This is a very difficult and non-trivial task. Its solution is challenging and requires the use of the most sophisticated analytical techniques. Among them, high-resolution mass spectrometry (HRMS) and 2D NMR spectroscopy certainly dominate [11,12]. The latter has obvious advantages, making it possible to identify various structures in the lignin macromolecule without preliminary degradation of the biopolymer [13]. However, the complexity of NMR spectra of lignin and strong overlap of spectral peaks even in 2D NMR experiments makes the “manual” interpretation of the data very difficult. In this regard, most of the works available in the literature are focused on the search and quantification of the known and most abundant structural units using expert analysis of individual spectra and neglect the identification of minor fragments. However, they can play a key role in a detailed classification of lignins and deep understanding of their transformations in various biological and technological processes. To overcome this problem, chemometric approaches to the data mining and treatment, primarily hierarchical cluster analysis (HCA), principal component analysis (PCA), and partial least squares projection to latent structure (PLS), should be implemented. In addition to being widely used in metabolomic studies [14], they have proven themselves useful for revealing differences between lignin preparations based on FT-IR spectra [15,16]. Recently, Lancefield et al. [17] reported the use of PCA and PLS for the FT-IR analysis of 54 lignin samples differing in origin and fractionation procedure. 

Surprisingly, there are still no data in the literature on the use of multivariate analysis (MVA) of 2D NMR data to study the structural differences of lignins. Nevertheless, this approach has already been successfully used to discriminate wines based on their polyphenolic composition [18], as well as identify differences between extracts of the poplar phloem and in the chemical composition of normal and tension wood [19,20]. This is primarily due to the complexity of processing 2D NMR spectra for their subsequent analysis using conventional MVA software, as well as difficulties in the subsequent extraction of information about specific compounds or structural fragments that contribute to certain differences. The latter factor is the reason why the majority of publications available in the literature in this field are limited to sample classification purposes. At the same time, methods for solving such problems are quite well known in the literature [21] and have been further developed for wood components by Hedenström et al. [19]. They proposed a procedure for MVA of frequency domain 2D NMR data without any need for peak picking or integration prior to analysis, where the loading’s plot can be visualized as pseudo-HSQC spectra to identify potential biomarkers. 

The aim of this study is to expand the scope of this approach for the classification of lignins by botanical origin and to determine the key substructures (biomarkers) for their differentiation by PCA and HCA analysis of 2D NMR (HSQC) spectra. As an example, preparations of dioxane lignin isolated from coniferous and deciduous wood, as well as herbaceous plants, were studied.

## 2. Results and Discussion

### 2.1. General PCA-Based Classification of Lignins 

Differences in the chemical composition and structure of 34 dioxane lignin preparations (Table 1), including 12 coniferous, 10 deciduous, and 12 grass lignin samples, were visualized using MVA of pre-processed ^1^H-^13^C HSQC spectra according to the procedure described in Section 3. In the case of spruce, pine, juniper, and larch, plant material samples obtained from different trees were used for isolation of lignin preparations to estimate the intra-species variability of the lignin structure. 

The obtained PCA results showed that the observed data variations can be described by a total of 19 principal components. Based on the residual variance curve, it was found that four of them are sufficient to describe 70% of the differences (Appendix A), while PC1, PC2, PC3, and PC4 account for 39.9, 16.4, 8.5, and 6.0% of the total variance, respectively. The score plot for PC1 versus PC2 (Figure 1a) clearly demonstrates the distribution of the studied lignins into the three distinct clusters along the PC1 axis. The accumulation of points belonging to deciduous (hardwoods) lignins on the right side (large positive values of PC1) of the graph is observed, while the points on the left side (large negative values of PC1) correspond to coniferous (softwoods) lignins. The third cluster, related mainly to grass (herbs) lignins, is located near the center with a shift towards lignins isolated from hardwood samples. Replacing PC2 on the *y*-axis with PC3 made it possible to better differentiate grass lignins highlighting the areas corresponding to monocots and dicotyledons (Figure 1b). The score plot in the PC1-PC4 coordinates (Figure 1c) provided a complete separation of the lignins isolated from deciduous hardwood and grass (herbs). 

The loading plots illustrating in detail the key structures responsible for the differences between the clusters in four principal components (PC1-PC4) were converted back to the corresponding HSQC NMR spectra (Appendix A), which allowed the main structures responsible for differences between the studied lignins to be revealed (Table 2).

As expected, the main differences between lignins are in the ratio of syringyl and guaiacyl monomeric units (S/G), namely, in the presence of S-structures in the composition of deciduous lignins, and vice versa, the predominance of G-structures in coniferous lignins. In the case of herbaceous lignins, the important role is played by the presence of H-type structures, which make a significant contribution to PC2. However, the composition of the main dimeric structures also largely affects the differentiation of lignins. In particular, fragments of β-aryl ether and resinol predominate in the composition of deciduous lignins, while the substructures of phenylcoumarane and secoisolariciresinol types are characteristic of coniferous lignin. This is in good agreement with the literature data. It is worth noting that the signals belonging to Hibbert’s ketone and its isomers are observed as negative peaks for PC1, PC2, and PC3 and thus dominate in the composition of coniferous lignins. This may indicate that the destruction of β-O-4 bonds in them during the isolation procedure takes place to a greater extent.

The differentiation of grass lignins is not so unambiguous. In particular, the dioxan-lignin of the dicotyledonous plant Sosnowsky’s hogweed (*Heracleum*) is located in the cluster of lignins isolated from hardwood. This may indicate that its chemical composition is distinguished with a significant proportion of S-structures and the absence of cinnamic acids which are characteristic of grass lignins. Another illustrative example is the lignin of saxifrage [22], which is located separately on score plots in PC1-PC2 and PC1-PC3 coordinates and goes beyond the confidence interval of herbaceous lignins area. This is due to the intense signals belonging to the H-units, fatty acids, and acetylated β-aryl ethers (Appendix A). 

Based on the data presented in Appendix A, it can be seen that the cross-peaks of the flavonoid tricin, *p*-coumaric, and ferulic acids, as well as arabinofuranose, make the greatest contribution to the differentiation of grass lignins along the PC3 axis. The latter compound forms an ester bond with ferulic acid and along with tricin can be considered a main biomarker of lignins in cereal straw. 

In the aforementioned differentiation of lignins of monocotyledonous and dicotyledonous herbaceous plants in the PC1-PC3 coordinates (Figure 1b), cattail (*Typha*) lignin which unexpectedly falls into the zone of dicotyledonous plants can be considered an exception. This is explained by the absence in its structure of flavonoid tricin fragments characteristic of monocots [23]. It is also noteworthy that some lignins isolated from dicotyledonous grasses are close to the cluster of deciduous lignins, which indicates a substantial similarity in their structure.

The combination of PC1 and PC4 for constructing the score plot (Figure 1c) proved to be most suitable to distinguish herbaceous and deciduous lignins. This suggests that the main part of the PC4 is contributed by the signals of substructures responsible for this classification. As can be seen from the loading plot HSQC spectrum (Appendix A), the cross-peaks related to S/G/H monomeric units have no decisive effect on the separation of lignins along the PC4 axis, and the most intense contours are observed for signals which are characteristic of dimeric structures and some of their degradation products.

### 2.2. PCA-Based Discrimination of Coniferous Lignins

The results described above show that the coniferous lignins not only differ significantly from the lignins of hardwood and herbaceous plants but are also characterized by a significantly lower structural variability compared to the latter. In this regard, the differences in the composition and structure of dioxane lignins of coniferous trees belonging to different species were analyzed in more detail. The results of the PCA showed that 100% of the variation in the data can be described by 11 principal components, of which PC1 and PC2 account for 29.3 and 21.7%, respectively (Appendix A). Combining the data obtained from the score plots and loading spectra, it was possible to establish the particular substructures which are characteristic of each species under study (spruce, pine, juniper, larch). 

Obviously, when using the score plot in PC1-PC2 coordinates (Figure 2a), a clear separation is observed only between the spruce lignin, which is distinguished with highly positive PC1 values, and all other preparations. For the latter, intraspecies variations turned out to be comparable with inter-species differences. For example, pine and juniper lignins, despite belonging to different families, demonstrated very similar patterns indicating the identity of their main substructures. The HSQC spectrum extracted from the loading plot (Appendix A) demonstrated that the main contribution to the special position of spruce lignin along the PC1 coordinate is made by Hibbert’s ketones, methyl-substituted phenylcoumarone, as well as the structures of vanillin and acetovanillone (Appendix A). On the other hand, cross peaks of H-type aromatic units, β-aryl ethers, and phenylcoumarane are characteristic of other coniferous lignins, the points of which are located on the left side of the score plot.

Based on the obtained results, two reasons can be suggested to explain the observed picture. First of all, the absence of H-type structures in the composition of spruce lignins may be due to the fact that the age of the plants selected for the isolation of spruce lignin preparations was about 80 years, while other representatives of coniferous trees were aged 25–30 years. Another reason may be associated with inherent limitations of the method caused by the lignin isolation procedure. In the latter, hydrochloric acid solution (0.7%) in dioxane was used as a mild hydrolytic agent facilitating the release of lignin. However, it also contributes to undesirable side processes of acid-catalyzed destruction and transformation of most labile structures in biopolymer macromolecules. They result in the formation of larger amounts of Hibbert’s ketones and other degradation products. Thus, to avoid possible classification errors, it is necessary to know which of the detected structures belong to intact lignin and which were formed during its isolation from a plant material. In the case of differentiation of lignins by classes, this factor may not have a significant effect, while a more detailed analysis, such as classification of coniferous lignins by families, is greatly complicated.

Of all the samples, only larch lignin preparations were not grouped together. An analysis of the PC2 loading HSQC spectrum (Appendix A) makes it possible to establish the reason for such a strong displacement of the Larch 1 sample, which is located on the border of the confidence ellipse on the left side of the PC2 axis. The key structural differences in this case are due to the presence in this sample of taxifolin fragments not observed in other lignin preparations. The dramatic differences between individual lignin samples within the same tree species may be associated with the predominant contribution of random impurities in PC1-PC2, as well as the phasing error of some cross peaks [19], likely associated with residual solvents, which were not completely removed during the pre-processing of the spectra. In addition, these differences may be explained by the side reactions in the delignification process such as the destruction of the β-O-4 bond under mild conditions of acidolysis [24]. However, as previously noted in [16], the relative amounts of degradation products depend on the chemical composition of plant material, which in turn depends on the plant species.

The complete separation of the pine, larch, and juniper lignin areas on the score plots was not achieved either in the PC1-PC2 coordinates, or even when using other principal components with the highest contribution (up to PC6). This means that lignins of coniferous tree species have a fairly similar structure; therefore, a distinctive feature for each of the lignins may not be significant for the tested principal components. To this end, other PCs with less contribution to the description of all variations were analyzed revealing more subtle differences in the biopolymer structure. As a result, it was found that the signals described by PC7 (Figure 2b) may be partially responsible for the clustering of softwood lignins by species (families). In particular, they provide clear separation of juniper lignin from the preparations isolated from larch and pine.

The data obtained from the loading spectrum of PC7 (Appendix A) showed that the distinguishing feature of juniper lignin is the absence of divanillyltetrahydrofuran and secoisolariciresinol fragments in their structure, as well as a large proportion of β-O-4 bonds and Hibbert’s ketones. In turn, pine lignin samples located in the area at the low PC7 values contained the mentioned structures and their composition is dominated by fragments of dibenzodioxocin, methyl-substituted phenylcoumarone, and other degradation fragments, including vanillin and vanillic acid (Appendix A).

Summarizing the results of PCA analysis and structural information from HSQC NMR loading spectra of lignins of various origins, the following classification of the studied biopolymer preparations presented in Figure 3 as a block scheme [25] can be proposed based on the specific substructures (biomarkers) identified in them.

### 2.3. Hierarchical Clustering of Lignins

In general, HCA analysis of the 2D HSQC NMR spectra of the studied lignins showed the same pattern as PCA. However, it allowed for a more detailed and clearer clustering of preparations isolated from the plants of various species and families (Figure 4). 

Two large clusters are observed on the dendrogram, one of which, in turn, is divided into two separate subclusters of deciduous and herbaceous lignins. Another subcluster involves mainly coniferous lignin preparations. This differentiation is obviously caused by the ratio of the main types of PPU. It should also be noted that 3 of the 34 analyzed samples were differentiated separately. Two of them belong to cereal straw lignin, which is explained by the presence of flavonoid-type substructures and other impurities in them. In addition, it was shown that aspen lignin is not a part of the hardwood lignin subcluster, which is explained by the presence in its structure of fragments that were largely modified during isolation procedure.

In addition, it can be seen that the differentiation between the lignins of dicotyledonous and monocotyledonous grasses was not clear enough. This is evidenced by the clustering of wheat and cattail lignins together with representatives of dicotyledonous grass lignins. However, subclustering of lignins by families within the cluster of hardwood lignins attracted the most attention. It is known that most of them have a similar composition of substructures, differing only in their quantitative ratio. The exceptions are willow, aspen, and poplar lignins, which contain structures of *p*-hydroxybenzoates (Figure 3). The cluster of coniferous lignins also undergoes differentiation, but the differences in this case are not so predictable and require a more detailed study.

## 3. Materials and Methods

### 3.1. Plant Material and Dioxane Lignin Isolation

The samples of saxifrage (*Saxifraga oppositifolia* L.) stems were obtained from Piramida settlement (Svalbard, Norway). Other plants were harvested in the Primorskii district of the Arkhangelsk region (Russia). At least three samples of each plant species from different sites were averaged prior to lignin isolation. Dioxane lignins were isolated from plant tissues (xylem of woody plants, grass aerial part, cereal straw) by the Pepper’s method [26], involving a mild acidolysis of lignocellulosic biomass in an inert atmosphere and extraction of lignin in water-dioxane medium. Before isolation of lignin, the plant biomass (xylem of woody plants, cereal stalks and aerial parts of other herbaceous plants) was crushed in a ZM 200 centrifugal mill (Retsch, Haan, Germany) to a particle size of <1 mm, vacuum dried at 40 °C, and subjected to exhaustive extraction with acetone in a Soxhlet apparatus to remove low-molecular extractives. A complete list of the obtained lignin preparations, their elemental compositions and molecular weight characteristics, as well as the attained yields are presented in Table 1. Determination of the carbon and nitrogen content were carried on an elemental (CHNS) analyzer EA-3000 (EuroVector, Pavia, Italy). The calculation of the oxygen content was carried out by the difference. Data are reported in weight percent as the average of the three replicates. The mean square deviation of the random component of the measurement error was 0.3% for C and 0.1% for H. Number-average (M_n_) and weight-average (M_w_) molecular masses were determined by size-exclusion chromatography on an LC-20 chromatographic system (Shimadzu, Kyoto, Japan) consisted of an LC-20AD pump, a DGU-5A vacuum degasser, an STO-30A column thermostat, an SIL-30AC autosampler, and an SPD-M20A diode array UV-VIS spectrophotometric detector. The separation was carried out on at 40 °C on an MCX column, 300 × 8 mm, pore size 1000 Å (PSS, Mainz, Germany). Aqueous solution of sodium hydroxide (0.1 M) was used as a sample solvent and mobile phase. 

### 3.2. Sample Analysis Using 2D ^1^H-^13^C HSQC NMR

In total, 50–80 mg of the dry lignin powder was dissolved in 0.55 mL of DMSO-d_6_ (Deutero GmbH, Kastellaun, Germany) and transferred to a 5 mm NMR tube. The ^1^H-^13^C HSQC spectra were recorded on a Bruker AVANCE III 600 MHz spectrometer (Bruker Biospin, Rheinstetten, Germany) using the Bruker library hsqcedetgpsisp2.3 pulse sequence. The experiments were carried out at 298 K using the following acquisition parameters: size of FID—1024 (F2) and 256 (F1), number of scans—32, relaxation delay—2 s, spectral width—15 ppm (F2) and 239 ppm (F1), transmitter offset—5.719 ppm (F2) and 98.2 ppm (F1), ^1^J_C-H_ = 145 Hz. After zero filling, the resulting data matrix size was 1024 (F2) × 1024 (F1). The spectra were processed using Bruker’s Topspin software version 3.2. 

### 3.3. Spectral Processing and PCA/HCA Analysis

Despite the fact that the assignment of the main cross-peaks is well described in the literature, visual comparison of data and integration of peaks related to the main substructures does not give a complete picture and does not allow us to establish a specific set of fragments characteristic of a particular lignin. The solution to this problem is possible by analyzing the data array of 2D HSQC spectra using multidimensional analysis. In this work, a combination of PCA/HCA methods was used. An overview of the procedure is shown in Figure 5.

The initial data was the HSQC spectrum data matrix in the range δC/δH 0-210/0-10 ppm, which includes all cross-peaks characterizing the composition and structure of lignin preparations. Preparation and preprocessing of NMR spectra for multivariate analysis was performed in accordance with protocol described by Hedenström et al. [19]. The obtained spectral data were analyzed by the PCA in order to get the most complete picture of the differences between the spectra and, consequently, the differences in the composition and structure of lignin preparations. Based on the data of the residual dispersion curve, an optimal set of principal components was determined, which were subject to further analysis and the construction of a scores and loadings plots. The latter were presented in the format of pseudo-HSQC spectra according to Hedenström et al. [19]. The sets of cross-peaks observed on these spectra and their intensity allowed us to draw conclusions about the contributions of certain fragments to the classification of lignins.

The scores used to construct scores plots were used as a distance matrix, allowing us to confirm the clustering of the studied lignin samples using the hierarchical method (HCA).

POKY software (version 20220114) was used for preliminary modeling of HSQC spectra [27]. For multidimensional analysis, pre-processing of all 2D ^1^H-^13^C HSQC spectra was performed in MATLAB R2021a software (Mathworks Inc., Natick, MA, USA), as described in Ref. [19]. PCA analysis was performed by MarkerView software version 1.2.1 (ABSciex, Toronto, ON, Canada) using Pareto scaling. The choice of this scaling option is based on the fact that it preserves the shapes of spectral lines better when loading pseudo-spectra [14]. Subsequently, the load vectors containing spectral information strongly correlating with the main differences were converted back into a 2D NMR load spectrum using the MATLAB R2021a software. HCA was performed based on the data of the main components using the OriginPro 2019b software (OriginLab Corp., Northampton, MA, USA). The error rate for cross-validation of training data was 2.27%.

### 3.4. Lignin Substructure Assignments

Qualitative analysis of 2D HSQC NMR spectra of the studied lignins and pseudo-NMR spectra representing loading plots of the main components was carried out by comparison with data from the literature sources [5,9,22,28,29,30,31,32]. The correctness of the identification was confirmed by the presence of all characteristic cross-peaks of the indicated fragments and the compliance of their chemical shifts with the literature data. The maximum permissible deviation was at 1 and 5 ppm for ^1^H and ^13^C dimensions, respectively.

## 4. Conclusions

The use of 2D NMR spectroscopy in combination with PCA and HCA makes it possible to effectively classify lignins of various botanical origins at the level of belonging to classes and families of plants, and, in some cases, individual species. The reverse transformation of loading plots into the corresponding HSQC NMR loading spectra provides structural information about the latent components of lignin and its structural fragments that cause certain differences. As a result of the study of 34 coniferous, deciduous, and herbaceous lignins, 10 groups of main substructures that determine the differences between lignins were established. In addition to syringyl, guaiacyl, and *p*-hydroxyphenyl PPU, they include various terminal substructures: dihydroconiferyl alcohol, balanopholin, cinnamic acids, and tricin. It was shown that, in some cases, the substructures formed during the partial destruction of biopolymer macromolecules also have a significant effect on the classification of lignins of various origins.

To the best of our knowledge, this study represents the first attempt to implement 2D NMR spectroscopy in the detailed classification of lignins and identification of related biomarkers in their structure using MVA. Along with the demonstrated advantages of the proposed approach, it is necessary to note its natural limitations revealed in our study. They are primarily related to the difficulty of interpretation and assignment of signals in 2D NMR spectra and the presence of overlapping peaks. In our opinion, overcoming this problem is possible by combining 2D NMR and high-resolution mass-spectrometry data, which may be a part of future research. It should also be focused on the introduction of the PLS-based approaches and expansion of the range of studied lignins, including the technical preparations obtained during the industrial processing of biomass. 

## Figures and Tables

**Figure 1 ijms-24-12403-f001:**
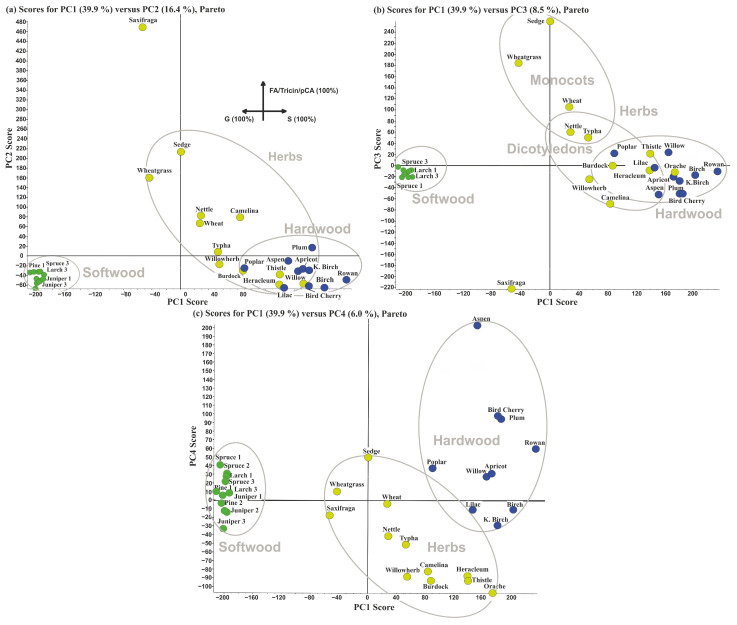
Score plots of PC2 (**a**), PC3 (**b**), and PC4 (**c**) versus PC1 as a result of PCA analysis of ^1^H-^13^C HSQC-NMR spectra of the studied lignin preparations: softwood lignin (green circles); hardwood lignin (blue circles); herbs lignin (yellow circles).

**Figure 2 ijms-24-12403-f002:**
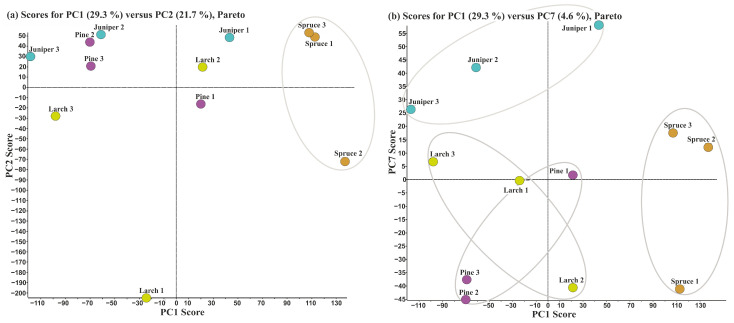
Score plots of PC2 (**a**) and PC7 (**b**) versus PC1 as a result of PCA analysis of ^1^H-^13^C HSQC-NMR spectra of the coniferous lignin preparations: juniper lignin (light blue circles); larch lignin (yellow circles); pine lignin (purple circles); spruce lignin (orange circles).

**Figure 3 ijms-24-12403-f003:**
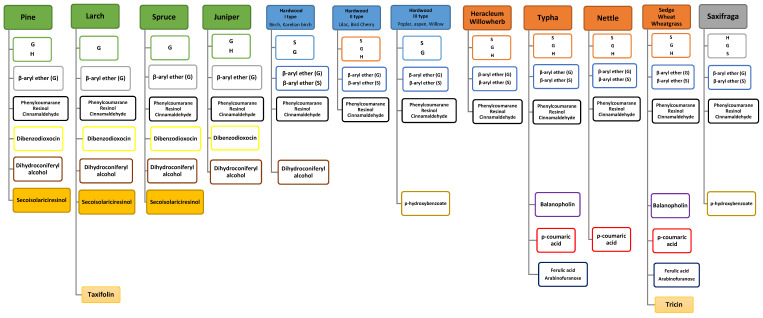
Block diagram of key substructures obtained as a result of the analysis of PCA-based HSQC NMR loading spectra of lignins of various biological origins.

**Figure 4 ijms-24-12403-f004:**
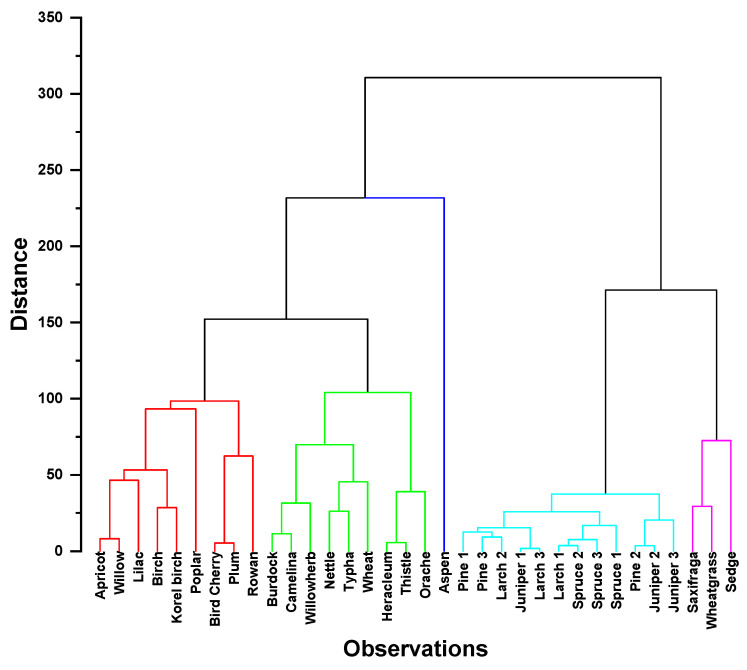
Dendrogram obtained by hierarchical cluster analysis of 2D HSQC NMR spectra of the studied lignins of various biological origins: hardwood lignin (red lines); herbs lignin (green lines); softwood lignin (light blue lines); lignin containing flavonoids in its structure (purple lines).

**Figure 5 ijms-24-12403-f005:**
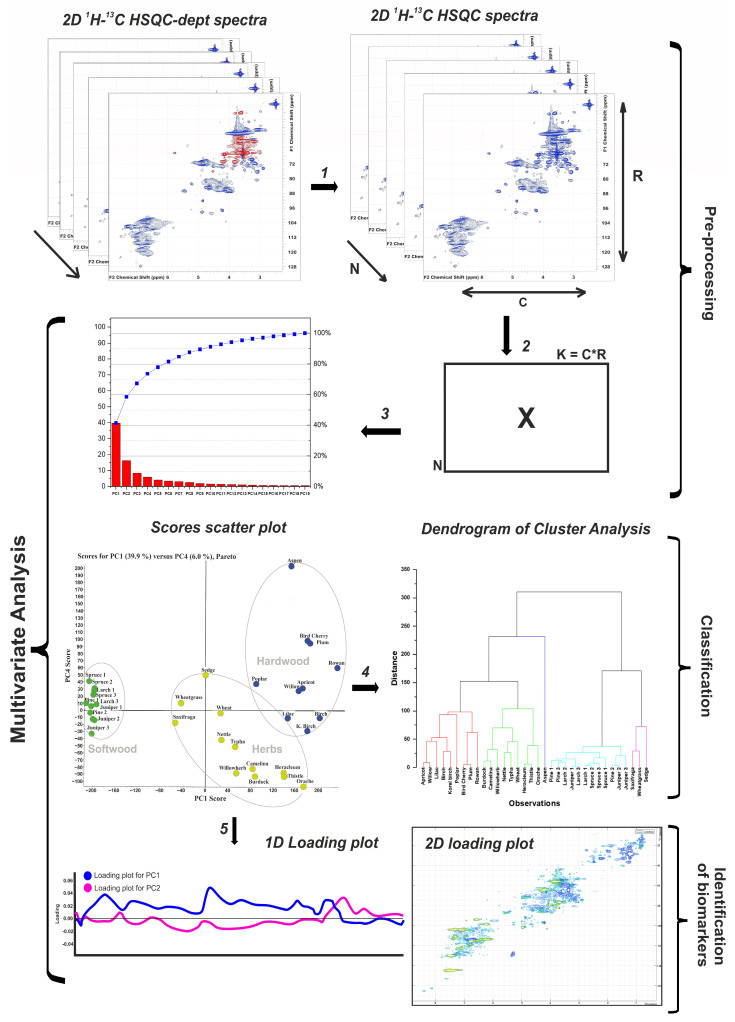
Overview of the procedure for multivariate analysis of 2D NMR data: (1) Each spectrum was processed by the POKY software in order to change the phase of negative cross-peaks to positive. (2) Each pre-processing spectrum is converted to a row vector and placed in a new data matrix X described in [19]. (3) Scores and loadings resulting from multivariate analysis of matrix X are performed. (4) Data from scores plot are used to hierarchical cluster analysis. (5) The loadings, initially represented as line plots, are converted to 2D loading spectra by reversing the unfolding procedure described in (2).

**Table 1 ijms-24-12403-t001:** Lignin preparations and their main characteristics.

Plant	Family	Yield, %	Elemental Composition, %			Molecular Weight, g mol^−1^		S/G/H Content, %			Main Substructures *, (Per 100 Aromatic Units)			
			C	O	H	M_n_	M_w_	S	G	H	A	B	C	D
Softwood														
Spruce (*Picea abies*)	Pinaceae	15	67.0	26.6	6.4	3000	8400	0.6	99.0	0.3	13.5	8.3	4.1	2.1
		11	62.6	30.1	7.3	1800	4600	1.0	99.0	0.0	11.3	7.8	4.2	1.7
		12	61.9	30.6	7.5	2300	5200	0.8	98.9	0.3	12.2	7.9	4.0	1.9
Cedar pine (*Pinus cembra*)		10	64.2	29.0	6.8	980	4100	0.0	95.4	4.6	16.3	9.9	5.7	1.5
		8	63.8	28.4	7.8	950	3600	1.3	95.9	2.8	16.7	8.2	3.4	2.6
		10	63.6	28.7	7.7	1150	4300	1.4	95.3	3.2	16.0	8.5	3.4	2.9
Larch (*Larix sibirica*)		10	61.2	32.0	6.8	950	3800	0.8	98.8	0.4	20.8	8.5	3.4	3.1
		9	63.8	29.2	7.0	1300	4100	1.3	98.7	0.0	14.8	7.4	4.3	0.9
		6	61.7	31.1	7.2	1750	4600	2.1	93.6	4.3	19.8	9.3	3.2	3.3
Juniper (*Juníperus commúnis*)	Cupressaceae	8	63.0	30.0	7.0	2300	5800	1.1	97.3	1.6	13.4	7.9	3.6	1.3
		7	64.0	28.5	7.5	950	4200	0.8	96.5	2.7	17.4	8.7	3.6	2.0
		5	63.4	28.9	7.8	1600	4800	1.4	96.1	2.5	19.6	9.3	3.8	3.3
Hardwood														
Apricot (*Prúnus armeníaca*)	Rosaceae	10	57.2	35.5	7.3	2100	6800	79.3	17.9	2.8	30.3	1.7	10.4	6.0
Plum (*Prúnus doméstica*)		10	55.0	37.9	7.1	1900	7200	80.3	16.7	3.0	36.0	1.4	8.4	0.0
Bird cherry (*Prúnus pádus*)		11	56.8	35.8	7.4	1350	3700	66.0	33.8	0.2	34.6	2.8	9.8	23.4
Rowan (*Sórbus aucupária*)		10	55.7	37.0	7.3	740	5500	81.3	18.7	0.0	29.3	0.9	7.8	6.6
Birch (*Bétula pubéscens*)	Betulaceae	15	59.9	33.2	6.9	1700	4400	73.7	26.5	0.0	40.0	2.1	9.3	7.3
Karelian birch (*Betula pendula var. Carelica*)		10	57.9	34.2	7.9	730	3500	68.1	31.6	0.3	38.0	2.7	9.1	5.3
Lilac (*Syringa vulgaris*)	Oleaceae	7	59.7	32.6	7.7	1900	4800	64.2	35.7	0.2	32.3	3.3	9.0	5.0
Willow (*Sálix babylónica*)	Salicaceae	10	59.7	32.6	7.7	1020	5700	63.5	34.2	2.3	32.9	4.2	9.6	5.6
Aspen (*Pópulus trémula*)		15	57.6	32.0	8.6	680	2700	67.3	27.1	5.6	24.9	2.1	8.1	15.9
Poplar (*Populus alba*)		15	59.5	33.0	7.5	970	3400	46.8	32.5	20.8	29.7	3.1	3.9	3.4
Monocots														
Cattail (*Týpha latifólia*)	Typhaceae	5	58.2	33.9	7.9	2700	10,050	40.2	55.3	4.5	37.7	5.8	6.0	5.4
Sedge (*Cárex heleonastes*)	Cyperaceae	5	59.2	32.4	6.7	300	2000	41.4	42.8	15.9	24.6	3.3	2.4	0.0
Wheat (*Tríticum*)	Poaceae	4	59.5	30.9	9.6	780	3000	41.7	50.4	7.9	29.5	3.9	2.4	4.2
Wheatgrass (*Elytrígia répens*)		7	58.9	32.4	6.8	300	2050	29.8	56.7	13.5	25.1	4.8	2.0	0.0
Dicotyledons														
Hogweed Sosnowskyi (*Heracleum*)	Apiaceae	6	60.2	8.5	31.3	1800	3500	55.9	43.3	0.8	43.1	4.8	11.3	6.2
Willowherb (*Epilóbium*)	Onagraceae	4	59.4	33.5	7.1	1430	2890	37.3	37.0	25.8	23.7	2.6	0.0	0.0
Nettle (*Urtíca dióica*)	Urticaceae	4	60.3	37.5	2.2	1400	1700	34.5	55.3	10.2	31.9	7.0	7.2	3.7
Saxifraga (*Saxifrága oppositifólia*)	Saxifragaceae	0.4	60.5	32.1	7.4	1500	2500	14.0	48.6	37.5	22.5	3.8	0.0	0.0
Thistles (*Cárduus*)	Asteraceae	4	58.7	30.3	9.2	730	3530	58.1	37.9	4.0	39.6	4.0	11.5	2.9
Burdock (*Árctium*)		4	57.7	31.6	8.4	950	2800	50.7	48.4	0.9	39.7	4.2	8.5	3.4
False flax (*Camēlina*)	Brassicaceae	0.4	59.0	29.0	9.2	780	4730	50.2	46.9	3.0	35.3	5.6	11.6	2.7
Orache (*Atriplex*)	Amaranthaceae	2	57.3	32.6	8.5	390	2950	63.0	36.2	0.8	38.9	3.5	15.1	3.3

* A—β-aryl ether, B—phenylcoumarane, C—resinol, D—1,3-dioxane. Mn—molecular masses number-average, Mw—molecular masses weight-average.

**Table 2 ijms-24-12403-t002:** Substructures responsible for the main differences between lignins along PC1-PC4 axes (“+” and “−” denote positive and negative correlations, respectively).

Substructure	Label	PC1	PC2	PC3	PC4
Main substructures					
Syringyl PPU	S	+	−	−	−/+
Guaiacyl PPU	G	−	−	n/d *	−
*p*-Hydroxyphenyl PPU	H	n/d	+	n/d	−
*p*-hydroxybenzoate	pB	+	n/d	n/d	+
β-aryl ether (S)	A	+	−	n/d	−
β-aryl ether (G)		−	−	n/d	−
β-aryl ether (H)		+	−	n/d	n/d
Phenylcoumarane	B	−	−	+	−
Secoisolariciresinol	Sc	−	n/d	n/d	n/d
Dibenzodioxocin	D	−	n/d	n/d	n/d
Resinol	C	+	−	−	−
Dihydroconiferyl alcohol	DCA	−	−	−	−
Cinnamaldehyde	J	−	−	−	−
Ferulic acid	Fa	n/d	+	+	n/d
*p*-coumaric acid	pCA	n/d	+	+	n/d
Balanopholin	BF	n/d	+	+	−
Substructures formed during isolation					
Methyl substituted phenylcoumarone	P	−	−	n/d	+
3,4-divanylyltetrahydrofuran	Di	−	n/d	n/d	n/d
1,3-dioxane structure	1,3D	−			+
Hibbert ketone	Hk	−	−	−	n/d
α-hydroxypropiovanillone		−	n/d	n/d	+
Acetovanillone	AV	−	−	n/d	+
Vanillin	V	−	−	n/d	+
Other					
Sugars	Sugars	+	+	−	+
Fatty acids (+Acetate)	Fatty acids (Ac)	+	+	−	− (+)
Arabinofuranose	Ar	n/d	+	+	n/d
Tricin	T	n/d	+	+	+

* Not detected (n/d).

## Data Availability

The data presented in this study are available in the article and Appendix A.

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
