# Peer review of "Application of 2D NMR Spectroscopy in Combination with Chemometric Tools for Classification of Natural Lignins"

_ijms, 2023, doi:10.3390/ijms241512403_

Round 1

Reviewer 1 Report

Too complex sentences should be divided into at least two sentences to improve clarity and readability. For example,  "Physical-chemical properties of lignin and  thus its suitability for certain directions of processing are determined by the chemical  structure of macromolecules, which is formed as a result of enzymatic dehydrogenative  polymerization of p-coumaryl, coniferyl, and synapyl alcohols (monolignols) which act as  precursors for p-hydroxyphenyl (H), guaiacyl (G), and syringyl (S) phenylpropane structural units, respectively."
The quality of the Figure No.5 should be improved. 

How was the elemental composition (in Table 1)  determined with a precision of 0.1 %? Are the percentages given in atomic or weight %? What is the error of the method used?

Too long, complex sentences and grammatical errors are present in the manuscript.

Reviewer 2 Report

The complex composition of lignin poses challenges in determining structural differences between lignins from different sources. In the current manuscript, authors explored the classification of lignins based on their chemical structure, utilizing 2D NMR spectroscopy and multivariate analysis. The overall concept is intriguing, and the findings hold considerable potential significance. Therefore, with the following minor revisions, this paper can be accepted:

1.     The authors mentioned that replacing PC2 with PC3 in Figure 1b allowed for better differentiation of grass lignins, Could the authors elaborate a little bit on the specific characteristics or features of PC3 that contributed to this improved differentiation?

2.     How does the isolation procedure impact the substantial destruction of lignin macromolecules? Have the authors detected notable discrepancies in the classification of lignins based on the substructures generated during the partial degradation of biopolymer macromolecules?

3.     The separation of pine, larch, and juniper lignin areas was not completely achieved in the PCA analysis. It would be helpful if the authors could provide some information on the limitations and challenges associated with the separation of these lignin areas, both in the PC1-PC2 coordinates and when considering other principal components.

4.     The structural heterogeneity and intricate nature of lignin can make the interpretation and assignment of signals in 2D NMR spectra challenging. The presence of overlapping signals and broad peaks can complicate the analysis, leading to difficulties in accurately characterizing specific lignin substructures. Therefore, it is important to include a critical assessment of the current understanding in the conclusion, along with potential applications and limitations of the findings. This will provide a comprehensive overview of the research and its implications for future studies and practical applications.

5.     I recommend the authors should crosscheck the manuscript for typos. For instances: line 324 ‘HSQS-dept’ should be HSQC; line 309-310 - complete the sentence by citing the authors of Ref. 19. 

6.     Reference should be uniform and updated as per the IJMS Style. 

Reviewer 3 Report

[Abstract]

1.     It is better to define all abbreviations in the abstract first. Such as NMR, HSQC, and so on.

2.     Please add the accuracy and other analytical parameters in the abstract.

[Introduction]

1.     Being the second most abundant biopolymer in nature, This seems like a repetition from the abstract. Please care to modify the phrase/sentence. Please do the same to the rest of the repetitive sentence that has been used in the abstract.

2.     Please make more justification on the use of NMR spectra as the data basis for chemometrics. Is it possible to perform NMR analysis on bulk samples (not the pure sample)?

3.     In the second paragraph of the introduction. Please make more highlights on the use of chemometric approach. For example by citing this reference: https://narrax.org/main/article/view/80

[Results and Discussion]

1.     If (*) in Table 2 indicates “not detected”. What does ‘n/d’ mean then?

[Methods]

1.     Cross validation should be carried out.

2.     Number of repetitions used for each sample should be declare. Also, please identify where and how the lignin was collected.

Please check for possible grammatical errors.

Reviewer 4 Report

Authors have elegantly demonstrated how the combination of two-dimensional NMR techniques and chemometric tools can be applied for the classification of lignins. The paper is well written and results are supported by experimental data and the adopted strategy can find application in the analysis and classification of many other complex molecules. The manuscript can be accepted in this present form.

Reviewer 5 Report

The manuscript by Faleva et al described a strategy to classify lignins and related biomarkers using a combination of two-dimensional NMR spectroscopy and multivariate analysis of 1H13C HSQC spectra.

The 2D NMR spectrum are not processed correctly, the phasing is off in all the spectrum presented.

It is absolutely not clear from the manuscript how the identification of different components in the NMR spectrum were made.

There are many typos for example: page 4, line 325.

The quality of figures is not of publishable quality, and it is very hard to read figure 5 particularly.

NA

Round 2

Reviewer 3 Report

I have no further comments. Authors have made sufficient response.

None.

Author Response

Аuthors are grateful to the referee for contribution to the improvement of the manuscript.

Reviewer 5 Report

I am satisfied with the revision.

NA

Author Response

Authors are grateful to the referee for his contribution to the improvement of the manuscript.